# A Study on Representation Transfer for Few-Shot Learning

## Abstract

Few-shot classification aims to learn to classify new object categories well using only a few labeled examples. Transfering feature representations from other models is a popular approach for solving few-shot classification problems. In this work we perform a systematic study of various feature representations for few-shot classification, including representations learned from MAML, supervised classification, and several common self-supervised tasks. We find that learning from more complex tasks tend to give better representations for few-shot classification, and thus we propose the use of representations learned from multiple tasks for few-shot classification. Coupled with new tricks on feature selection and voting to handle the issue of small sample size, our direct transfer learning method offers performance comparable to state-of-art on several benchmark datasets.

## 1 Introduction

Few-shot learning(FSL) aims to train a prediction model with very few labeled examples. It is motivated both by our interest in the ability of humans to learn new concepts with very few examples (Carey & Bartlett, 2011), and a practical need to build machine learning models without the expensive process of collecting a large amount of labeled training data. In typical few-shot learning settings we assume we have access to a large amount of data (labeled or unlabeled) from a source domain as prior knowledge, and a small amount of labeled data for a target task in a different domain.

Popular approaches for the few-shot learning problem include meta-learning and transfer learning. Meta-learning creates learning tasks that are similar to the target task from the source domain data, and tries to learn a model that can quickly adapt to a new task (Finn et al., 2017; Ren et al., 2018; Wang et al., 2019). Transfer learning, on the other hand, transfers models learned from the source domain data to the target task through different adaptation techniques. In this work we focus on transfer learning through representations, which aims to learn a good feature representation from source data so that when learning the target task we can focus on training a much simpler task model (e.g. a linear classifier) instead of having to learn feature extraction at the same time. In computer vision and natural language processing, pre-training with large amount of data has become the dominant method for building state-of-art models (Iglovikov & Shvets, 2018; Li et al., 2018; Cui et al., 2018; Ruder et al., 2019). For few-shot classification, recent works (Tian et al., 2020) show that even pre-training with simple classification task in the source domain gives a good feature extractor for training few-shot models.

In this work we attempt to systematically study different representations for few-shot learning to identify the good ones, including representations from common supervised and self-supervised pre-training tasks (Gidaris et al., 2018; Doersch et al., 2015; He et al., 2020; Chen et al., 2020a), and also representations from the meta-learning method MAML(Finn et al., 2017) by treating it as a feature extractor. Compared to typical evaluations of self-supervised representation learning which usually involve the same data distribution (train and test on same input data but for different tasks), few-shot learning differ in two important aspects. The first one is data shift: the training and test data have different distribution and usually contain different object categories. The second one is extreme small sample size compared to typical evaluations of transfer learning. These differences make a proper study of these different representations worthwhile.

In addition to learning good representations from source data, we believe there should also be emphasis on how to use them properly in few-shot settings. Representations learned from a large

amount of pre-training data are usually high-dimensional, and for some few-shot learning task (e.g., 5-way 1-shot classification) it is difficult to figure out which features are directly related to the target concept given only a handful of examples. In this work we also investigate ways to perform feature selection and noise reduction when adapting to the target task, given a good representation learned.

In summary, our contributions in this work include: 1) We study systematically different representations for few-shot classification, to understand why some representations work better than others. 2) We investigate the use of multi-task feature representations in few-shot learning and show their utility. 3) We introduce feature selection via training with auxiliary classes, and a voting scheme that follows naturally from multi-task prediction to improve few-shot learning.

## 2 RELATED WORKS

Motivated by bridging the gap between AI and human learning ability, we have seen an ever increasing attention in the area of few-shot learning in machine learning community. Model Agnostic Meta-Learning (MAML) was proposed by Finn et al. (2017); it aims to search for a proper initialization of the neural network that can be rapidly adapted to variety of new tasks with a few optimization steps. Many improvements have been proposed based on MAML including first-order approximations (Nichol et al., 2018) or incorporating task-specific information (Lee & Choi, 2018), etc. Inspired by the concept of "learning to learn", many metric-learning methods are also developed via meta-learning. By simulating the few-shot task setting while training the backbone model, these metric-learning models (i.e Prototypical Networks, Relation Networks) (Snell et al., 2017; Sung et al., 2018) aim to learn the best feature representation which can be generalized well to unseen novel tasks.

Our work is motivated by the recent studies Chen et al. (2019) and Tian et al. (2020), which show that pre-training a good feature extractor (e.g., using supervised classification) gives very good performance on few-shot classification. We continue along this line of study to investigate which pretraining tasks give the best feature extractor for few-shot classification. Our study of MAML as a feature extractor is motivated by the work in Raghu et al. (2019), which show that MAML in few-shot learning benefits more from the learned features rather than fast model adaptation.

Gidaris et al. (2019) and Su et al. (2020) study the use of self-supervised tasks including rotation(Gidaris et al., 2018) and jigsaw(Noroozi & Favaro, 2016) to improve few-shot learning. They use the self-supervised tasks as feature regularizers on top of meta-learning algorithms such as ProtoNet(Snell et al., 2017) and MAML(Finn et al., 2017), which is different from our direct joint representation learning followed by transfer. Our approach is also closely related to Doersch & Zisserman (2017) which consider jointly using multiple self-supervised tasks to learn visual representations for classification. Zhai et al. (2019) has done a large scale study of the performance of representation learning on a large set of downstream tasks, while we focus on few-shot classification in this work.

## 3 REPRESENTATION TRANSFER FOR FEW-SHOT LEARNING

In all the following experiments we adopt the following protocol to investigate transfer learning. We first train a feature representation using one or multiple tasks on the source domain data, and use the last layer of activations as our feature representation. During adaptation stage to the new fewshot domain, we train a linear logistic regression classifier on top of the fixed feature representation to adapt the model. This is a common protocol in many evaluations of representation learning methods. In this work we focus on two backbones: ConvNet4 and ResNet12. ConvNet4 is a 4-layer convolutional neural network with hidden layer size of 64 in each layer, where each layer is a 3x3 convolution followed by ReLU, batchnorm, and max-pooling of stride 2. ResNet12 is a residual network (He et al., 2016) with 4 residual blocks of width 64-160-320-640. We also consider a wider version of ResNet12 which we denote as ResNetW12 by increasing the width to 128-320-640-640, to better accommodate multi-task learning. The feature representation size is fixed at 640. Our code is implemented in PyTorch (Paszke et al., 2019). We focus on the two most commonly used datasets in studying few-shot learning, Mini-ImageNet(Vinyals et al., 2016) and Tierd-ImageNet(Ren et al., 2018), in the work below. We found that there are two ways to construct the 84x84 inputs of these datasets from ImageNet (see Appendix A.1). We use the approach of directly taking random

resized crops of 84x84 on the ImageNet images, as it allows us to experiment on more diverse set of representation learning methods.

## 3.1 REPRESENTATIONS FROM META-LEARNING

We begin our study of representation transfer for few-shot learning with feature representations produced by MAML. Previous studies Raghu et al. (2019) show that the benefits of MAML come mainly from the feature representation learned in the last layer of the neural network instead of the ability to do fast adaptation. In this section we want to go further and consider the question: from a representation transfer point of view, what source tasks are the best for a target distribution of few-shot learning tasks? Is learning from a set of 5-way 1-shot source tasks always the best if we want to perform 5-way 1-shot classification in the target domain?

We use a modified version of MAML algorithm(Algorithm 1) that only adapts the last layer of the neural network on the support set, as adapting the last layer only performs very close to adapting the whole network (the almost-no-inner-loop(ANIL) algorithm in (Raghu et al., 2019)). We split the parameters of the neural network into weights for the linear layer $\phi$ and the rest of the earlier feature extraction layers $\theta$, and write $g_\phi(f_\theta(x))$ for the output of network with input $x$. We use $(X_i^S, Y_i^S, X_i^Q, Y_i^Q)$ denoting the support set inputs, support set labels, query set inputs, query set labels to represent the sampled few-shot classification task $\mathcal{T}_i$. Note that only the linear layer parameters $\phi$ are updated in the inner loop using the support set, reflecting that we only update the linear layer during adaptation. Parameters for feature representation $\theta$ are only updated in the outer loop.

---

**Algorithm 1** Modified MAML for few-shot classification that adapts only the last layer (ANIL)

---

Given: inner and outer step size hyperparameters $\alpha$, $\beta$, task distribution $p$
Randomly initialize $\theta$, $\phi$
**while** not done **do**
    Sample batch of tasks $\mathcal{T}_i \sim p(\mathcal{T})$
    **for** all $\mathcal{T}_i = (X_i^S, Y_i^S, X_i^Q, Y_i^Q)$ **do**
        Update $\phi_i' \leftarrow \phi - \alpha\nabla_\phi L(g_\phi(f_\theta(X_i^S)), Y_i^S)$
    **end for**
    Update $\theta \leftarrow \theta - \beta\nabla_\theta \sum_{\mathcal{T}_i} L(g_{\phi_i'}(f_\theta(X_i^Q)), Y_i^Q)$,
           $\phi \leftarrow \phi - \beta\nabla_\phi \sum_{\mathcal{T}_i} L(g_{\phi_i'}(f_\theta(X_i^Q)), Y_i^Q)$
**end while**

---

We train 5-way 1-shot(5w1s), 5-way 5-shot(5w5s), 10-way 1-shot(10w1s) and 10-way 5-shot(10w5s) tasks on Mini-ImageNet and Tiered-ImageNet, using both the ConvNet4 and ResNet12 architecture. To evaluate the representations learned, we extract the last layer features from these $n$-way $k$-shot networks, and train a logistic regression model on top of it during the adaptation stage on the support set. Each accuracy number reported in Tables 1 and 2 are the median of 5 trials, each of which is the average of 600 random draws of $n$-way $k$-shot tasks.

The MAML algorithm is implemented using the `higher` package (Grefenstette et al., 2019). We use an outer loop step size of 0.001 and inner step size of 0.01 for 5-way tasks and 0.05 for 10-way tasks. There are 5 inner loop steps during training and 10 inner loop steps during evaluations. We use a task batch size of 4 and run for 400 epochs. Due to the occassional overfitting of ResNet12 models, we select the model that performs best in terms of loss on the validation set provided in Mini-ImageNet and Tiered-Imagenet, and report the results on the test set.

From the results on ConvNet4 in Table 1, we can see that contrary to intuition, training and testing on the same type of $n$-way $k$-shot tasks do not result in the best performance. In general, representations learned from 10-way tasks perform better than representations learned from 5-way tasks, irrespective of the number of ways in the target tasks. And learning with more shots are also usually better. The results for ResNet12 are similar (Table 2), except for the 5w1s outlier in Mini-ImageNet. Snell et al. (2017) has previously noted the effect of models trained on larger number of ways performing better than models trained on number of ways matching the target task, while Cao et al. (2019) and Triantafillou et al. (2020) have studied the effect of number of shots in few-shot learning.

Table 1: Few-shot classification accuracies of different MAML source representations based on ConvNet4 on Mini-ImageNet and Tiered-ImageNet.

| | Mini-ImageNet | | | | Tiered-ImageNet | | | |
|---|---|---|---|---|---|---|---|---|
| source rep./target task | 5w1s | 5w5s | 10w1s | 10w5s | 5w1s | 5w5s | 10w1s | 10w5s |
| ConvNet4-5w1s | 49.25 (0.76) | 62.30 (0.67) | 32.90 (0.43) | 46.20 (0.42) | 48.95 (0.89) | 62.71 (0.77) | 33.71 (0.56) | 47.23 (0.52) |
| ConvNet4-5w5s | 51.07 (0.79) | 64.88 (0.72) | 34.85 (0.47) | 49.20 (0.44) | 51.91 (0.84) | 66.29 (0.76) | 36.52 (0.58) | 51.12 (0.56) |
| ConvNet4-10w1s | 51.68 (0.82) | 64.04 (0.73) | 35.12 (0.45) | 47.94 (0.43) | 50.98 (0.89) | 65.38 (0.77) | 35.55 (0.54) | 49.69 (0.52) |
| ConvNet4-10w5s | 49.71 (0.82) | 62.57 (0.73) | 33.57 (0.45) | 48.66 (0.43) | 52.54 (0.86) | 67.02 (0.78) | 37.11 (0.58) | 52.21 (0.55) |

Table 2: Few-shot classification accuracies of different MAML source representations based on ResNet12 on Mini-ImageNet and Tiered-ImageNet.

| | Mini-ImageNet | | | | Tiered-ImageNet | | | |
|---|---|---|---|---|---|---|---|---|
| source rep./target task | 5w1s | 5w5s | 10w1s | 10w5s | 5w1s | 5w5s | 10w1s | 10w5s |
| ResNet12-5w1s | 57.26 (0.84) | 69.51 (0.66) | 39.47 (0.51) | 52.83 (0.46) | 60.77 (0.95) | 72.87 (0.84) | 44.33 (0.65) | 58.58 (0.57) |
| ResNet12-5w5s | 53.78 (0.88) | 67.28 (0.70) | 36.63 (0.51) | 51.31 (0.45) | 61.93 (0.98) | 75.18 (0.75) | 46.00 (0.67) | 60.89 (0.58) |
| ResNet12-10w1s | 54.67 (0.81) | 68.39 (0.69) | 37.70 (0.54) | 52.46 (0.47) | 62.56 (0.97) | 75.52 (0.75) | 46.71 (0.64) | 61.27 (0.59) |
| ResNet12-10w5s | 53.86 (0.84) | 67.61 (0.67) | 36.83 (0.52) | 51.92 (0.44) | 63.98 (0.97) | 77.47 (0.78) | 48.78 (0.68) | 64.62 (0.57) |

From an adaptation point of view, 5-way models should transfer better to 5-way tasks, since they come from a more 'similar' task distribution. However, from a feature learning perspective, this is not surprising. More difficult pairing of classes like cat VS tiger are more common for higher way tasks in a random sampling setting, thus they force the models to learn more discriminative features that might transfer better to new domains.

Note that our adaptation here is based on logistic regression, which is different from fast adaptation by performing a few gradient steps in MAML using the linear weights learned in the last layer. The logistic regression results sometimes can be 1-2% lower than adaptation using MAML, since the feature representations are trained using MAML and the last layer linear weights also encode some prior information on what the adapted weights should be close to. However as we use logistic regression for all model adaptations the comparison should be fair, and the results indicate increasing the number of ways allow better representations to be learned for transfer.

## 3.2 REPRESENTATIONS FROM SUPERVISED AND SELF-SUPERVISED TASKS

In the last section we study the feature representations learned by MAML on few-shot learning tasks, by considering MAML as a feature learning algorithm. In computer vision it is common to pre-train on ImageNet classification, and then adapt the representation learned to downstream tasks such as image segmentation or object detection (Iglovikov & Shvets, 2018; Li et al., 2018). In recent years there are also many works on using self-supervised tasks to pre-train computer vision and NLP models (Chen et al., 2020a;b; Baevski et al., 2019; Lan et al., 2019). The paper Tian et al. (2020) showed that by just pre-training on the classification task using the source training set and adapting using logistic regression, they can obtain results in few-shot learning that are competitive with state-of-art algorithms. They also studied the use of distillation to improve the pre-training. In this section we follow this line of investigation and study systematically the few-shot learning performance of different supervised and self-supervised tasks.

We train these classification models using a learning rate of 0.05 for Mini-ImageNet and 0.01 for Tiered-ImageNet, for 100 epochs using batch size of 64. The learning rates are decayed by a factor of 10 at the 60th and 80th epochs. From the training partition of these two datasets, we further split the partition into a training set (consisting of 90% of training partition) for training the classification models, and a validation (remaining 10% of partition) for evaluation and model selection. As we can see from both Mini-ImageNet and Tiered-ImageNet in Table 3, adapting from classification repre-

Table 3: Few-shot classification accuracies of classification representation compared to MAML.

| source rep./target task | Mini-ImageNet | | | | Tiered-ImageNet | | | |
|---|---|---|---|---|---|---|---|---|
| | 5w1s | 5w5s | 10w1s | 10w5s | 5w1s | 5w5s | 10w1s | 10w5s |
| ConvNet4-class | 45.41 (0.75) | 62.38 (0.74) | 31.52 (0.45) | 47.35 (0.43) | 49.27 (0.85) | 66.28 (0.77) | 35.07 (0.51) | 52.07 (0.56) |
| ConvNet4-maml | 48.72 (0.79) | 64.88 (0.71) | 35.87 (0.43) | 47.26 (0.42) | 48.77 (0.85) | 65.65 (0.76) | 35.19 (0.51) | 51.30 (0.53) |
| ResNetW12-class | 61.75 (0.79) | 78.98 (0.60) | 47.09 (0.53) | 66.93 (0.44) | 71.36 (0.91) | 85.56 (0.64) | 58.15 (0.66) | 75.77 (0.54) |
| ResNet12-maml | 57.74 (0.76) | 65.97 (0.67) | 37.79 (0.47) | 51.94 (0.44) | 60.47 (0.84) | 74.69 (0.81) | 45.08 (0.52) | 63.49 (0.52) |

Table 4: Few-shot classification accuracies of different self-supervised source representations.

| source rep./target task | Mini-ImageNet | | | | Tiered-ImageNet | | | |
|---|---|---|---|---|---|---|---|---|
| | 5w1s | 5w5s | 10w1s | 10w5s | 5w1s | 5w5s | 10w1s | 10w5s |
| ResNetW12-rot | 34.61 (0.64) | 48.60 (0.66) | 22.10 (0.36) | 34.05 (0.40) | 35.40 (0.69) | 46.87 (0.69) | 22.10 (0.36) | 31.24 (0.40) |
| ResNetW12-loc | 32.86 (0.63) | 44.93 (0.65) | 19.92 (0.35) | 29.40 (0.38) | 26.17 (0.51) | 33.39 (0.59) | 14.75 (0.26) | 20.66 (0.34) |
| ResNetW12-contrast | 46.63 (0.75) | 64.83 (0.66) | 33.56 (0.48) | 50.75 (0.47) | 52.63 (0.86) | 72.00 (0.72) | 39.26 (0.58) | 59.37 (0.58) |

sentation using logistic regression does not beat training with MAML when we use the simpler CNN model ConvNet4. The main reason for this is the ConvNet4 model does not have sufficient capacity for the classification task (only 51% top-1 accuracy on Mini-Imagenet and 26% top-1 accuracy on Tiered-ImageNet), and thus the representation learned is not good enough for transfer. When we switch to the higher capacity ResNet12 model (81% top-1 for mini-ImageNet and 70% top-1 for tiered-ImageNet), the transfer learning approach works much better than MAML, with few-shot learning accuracies close to many state-of-art methods (see comparison methods in Table 11). This is consistent with the observations in Tian et al. (2020), which we are reproducing here for comparisons with other feature representations that follow. We believe the classification tasks (64 classes for Mini-ImageNet and 351 for Tiered-ImageNet) force the CNN models to learn richer and more stable feature representations than the 5-way or 10-way MAML classification tasks. It is uncommon to sample more difficult pairs of classes (e.g. cat VS tigers) in 5-way or 10-way MAML setup based on random sampling, and these difficult pairs make the models learn more distinguishing features.

In recent years there have been many studies on using self-supervision to pre-train classification models to reduce the requirement of labeled data (Gidaris et al., 2018; Doersch et al., 2015; He et al., 2020; Chen et al., 2020a). Below we also consider several common self-supervision pre-training tasks on FSL, including rotation prediction, location prediction (related to jigsaw), and contrastive learning, to see how well these self-supervised representations transfer to FSL tasks.

For rotation prediction (Gidaris et al., 2018), we randomly rotate the 84x84 input image 0, 90, 180, or 270 degrees and use these rotation angles as class labels. For location prediction (Sun et al., 2019), we sample the input image at a higher 168x168 resolution, and split the image into 4 equal parts (top-left, top-right, lower-left, lower-right), and use the location of the split as class labels. For contrastive prediction, we follow the same implementation as in the simCLR paper (Chen et al., 2020a), using an output embedding size of 128, and a batch size of 128. We tried larger batch sizes like 256 and 512 but this did not improve the results. These models are trained using step size of 0.05 and batch size 64 (apart from the contrastive models).

From Table 4 we can see that the self-supervised tasks provide fairly good feature representations for few-shot learning on Mini-ImageNet and Tiered-ImageNet. The relative strength of each feature representation is consistent with previous results on self-supervised learning with ImageNet (He et al., 2020), with rotation slightly better than location prediction and the more recent contrastive learning better than both of them. However, they are still not as good as direct transfer using the supervised classification training (Table 4).

To evaluate the properties of these self-supervised tasks further, we cross-evaluate the representations learned from one task on the other, using the 10% validation set from the training partition

Table 5: Prediction accuracies for holdout-evaluation of different self-supervised representations.

| train on/evalute on | Mini-ImageNet - holdout evaluation by task | | |
|---|---|---|---|
| | classification | rotation | location |
| classification | 81.33 | 47.14 | 53.27 |
| rotation | 30.16 | 84.64 | 59.00 |
| location | 22.79 | 43.13 | 79.80 |
| contrastive | 49.59 | 50.42 | 53.38 |

(which has the same distribution as these models are trained on). We just take the representations from these three models, and the re-train the last layer of the model using logistic regression using the 90% training data from the training partition (exact same data on which the representations are trained on), and test on the remaining 10%. We omit the contrastive representation because the contrastive loss is dependent on batch size and not easily comparable. We can see from Table 5 that the representations trained on one task usually perform reasonably on the other two, but is not competitive with models trained for that particular task. For example, the representation trained for classification does not perform particularly well for rotation (47.14%) and location (53.27%), both of which are 4-way classification problems. This motivates us to consider the question of whether we can train representations that excel in all these tasks, and whether such representations will be useful in few-shot learning.

## 4 MULTI-TASK REPRESENTATIONS

When training for feature representations separately, we minimize over the training data from the source domain the following objective functions:

$$\sum_{i=1}^{n} L_{\text{cls}}(f(x_i; \theta, w_{\text{cls}}), y_i^{\text{cls}}),$$
$$\sum_{i=1}^{n} L_{\text{rot}}(f(\mathcal{T}_{\text{rot}}(x_i; y_i^{\text{rot}}); \theta, w_{\text{rot}}), y_i^{\text{rot}}),$$
$$\sum_{i=1}^{n} L_{\text{loc}}(f(\mathcal{T}_{\text{loc}}(x_i; y_i^{\text{loc}}); \theta, w_{\text{loc}}), y_i^{\text{loc}}),$$

where $\mathcal{T}_{\text{rot}}$ and $\mathcal{T}_{\text{loc}}$ are the rotation and location-based cropping transform, and $L_{\text{cls}}$, $L_{\text{rot}}$, $L_{\text{loc}}$ are the corresponding cross-entropy loss for supervised classification, rotation prediction, and location prediction respectively. The neural networks are represented by $f(\cdot; \theta, w)$ with backbone parameters $\theta$ and linear(head) layer weights $w$.

To combine these different learning tasks into one shared neural network for joint training, a major difficulty is the differences in their inputs Doersch & Zisserman (2017). A rotated or a cropped input image can affect the learning of the classification task if the classification task is jointly trained with the rotation or location prediction task. Nevertheless, we consider jointly training the classification task with rotation or location prediction tasks with the transformed inputs:

$$\sum_{i=1}^{n} (L_{\text{cls}}(f(\mathcal{T}_{\text{rot}}(x_i; y_i^{\text{rot}}); \theta, w_{\text{cls}}), y_i^{\text{cls}}) + \lambda_{\text{rot}} L_{\text{rot}}(f(\mathcal{T}_{\text{rot}}(x_i; y_i^{\text{rot}}); \theta, w_{\text{rot}}), y_i^{\text{rot}})),$$
$$\sum_{i=1}^{n} (L_{\text{cls}}(f(\mathcal{T}_{\text{loc}}(x_i; y_i^{\text{loc}}); \theta, w_{\text{cls}}), y_i^{\text{cls}}) + \lambda_{\text{loc}} L_{\text{loc}}(f(\mathcal{T}_{\text{loc}}(x_i; y_i^{\text{loc}}); \theta, w_{\text{loc}}), y_i^{\text{loc}})),$$

where $\lambda_{\text{rot}}$ and $\lambda_{\text{loc}}$ control the relative weights of the tasks. The backbone parameters $\theta$ are shared among different tasks, while the weights $w_{\text{cls}}, w_{\text{rot}}, w_{\text{loc}}$ are task-specific heads.

We also consider jointly training all three tasks, based on the composed transformation $\mathcal{T}_{\text{loc+rot}}$, cropping based on location followed by random rotation.

$$\sum_{i=1}^{n} \left( L_{\text{cls}}(f(x_i'; \theta, w_{\text{cls}}), y_i^{\text{cls}}) + \lambda_{\text{rot}} L_{\text{rot}}(f(x_i'; \theta, w_{\text{rot}}), y_i^{\text{rot}}) + \lambda_{loc} L_{\text{loc}}(f(x_i'; \theta, w_{\text{loc}}), y_i^{\text{loc}}) \right),$$

where $x_i' = \mathcal{T}_{\text{loc+rot}}(x_i; y_i^{\text{rot}}, y_i^{\text{loc}})$.

We simply fix $\lambda_{\text{rot}}$ and $\lambda_{\text{loc}}$ at 1 (all three tasks have equal weights). We find that training with all 4 versions of a rotation or location crop in the same mini-batch works slightly better than randomly sampling one of them, although this can reduce the effective batch size for the supervised classification task. For Mini-ImageNet we adopt this approach of taking all 4 copies of rotation or

Table 6: Prediction accuracies for holdout-evaluation of different multi-task representations.

| train on/evalute on | Mini-ImageNet - holdout evaluation by task | | |
|---|---|---|---|
| | classification | rotation | location |
| ResNetW12-cls+rot | 81.59 | 86.35 | 60.75 |
| ResNetW12-cls+loc4 | 75.78 | 51.82 | 80.03 |
| ResNetW12-cls+loc5 | 80.86 | 52.37 | 75.21 |
| ResNetW12-cls+rot+loc5 | 78.31 | 85.94 | 72.74 |

Table 7: Few-shot classification accuracies of different multi-task trained representations.

| source rep./target task | Mini-ImageNet | | | | Tiered-ImageNet | | | |
|---|---|---|---|---|---|---|---|---|
| | 5w1s | 5w5s | 10w1s | 10w5s | 5w1s | 5w5s | 10w1s | 10w5s |
| ResNetW12 | 61.75 | 78.98 | 47.09 | 66.93 | 71.36 | 85.56 | 58.15 | 75.77 |
| -cls | (0.79) | (0.60) | (0.53) | (0.44) | (0.91) | (0.64) | (0.66) | (0.54) |
| ResNetW12 | 63.66 | 81.68 | 49.38 | 70.48 | 72.65 | 86.17 | 59.88 | 77.25 |
| -cls+rot | (0.80) | (0.57) | (0.54) | (0.42) | (0.86) | (0.61) | (0.65) | (0.51) |
| ResNetW12 | 61.67 | 79.19 | 46.89 | 67.33 | 70.55 | 85.33 | 57.33 | 75.43 |
| -cls+loc5 | (0.80) | (0.59) | (0.52) | (0.41) | (0.92) | (0.59) | (0.65) | (0.53) |
| ResNetW12 | 62.56 | 80.35 | 48.43 | 68.63 | 70.11 | 85.14 | 57.51 | 75.47 |
| -cls+rot+loc5 | (0.77) | (0.59) | (0.50) | (0.44) | (0.91) | (0.63) | (0.65) | (0.51) |

location crop in the same mini-batch, while for the larger Tiered-ImageNet we use the sampling version instead to reduce training time.

We again consider the holdout set accuracy on different tasks for these joint models. From Table 6 we can see that by jointly training with the other tasks, the performance on the corresponding task improves. For the joint model with location prediction (cls+loc4), we find that the classification accuracy on the holdout set drops by more than 5%, because for that model the training inputs are the 4 corners of an image and the model has never seen whole pictures of the object categories to be learned. We can see how the learning tasks can interfere with each other in this case. We therefore create a 5-class version of the location prediction task, by including the rescaled version of the whole picture as an extra class, in addition to the original 4 corners. The resulting model(cls+loc5) has improved classification accuracy. We can also observe that there is a tradeoff among accuracies of different tasks due to finite learning capacity of the models, and this is particularly evident in the joint model cls+rot+loc5.

From Table 7 we can see that for Mini-ImageNet, training a multi-task model with rotation prediction(cls+rot) significantly improves the few-shot learning accuracy of the representation. On the other hand there is no significant improvement with the model with joint location training. For the joint model with all 3 tasks (cls+rot+loc5), it is better than the base supervised representation but slightly worse than the joint model with rotation prediction only, indicating that there might be a capacity problem with fitting all three tasks.

For Tiered-ImageNet we again see an improvement with the multi-task representation with rotation prediction (Table 7). For the joint model with location and joint model with all 3 tasks (cls+rot+loc5) there are declines compared to the baseline model. The capacity constraint is more severe with Tiered-ImageNet as it is a more complex dataset with larger number of classes. The classification and location prediction accuracies are 60% and 89% on the holdout set for the cls+loc5 model, but once we add the rotation task the holdout accuracies for classification, location, and rotation prediction drop to 56%, 79% and 87% respectively.

## 5  MAKING USE OF AUXILIARY CLASSES AND VOTING AT ADAPTATION

### 5.1  IRRELEVANT FEATURE ELIMINATION WITH AUXILIARY CLASSES

In previous sections, we aim to investigate which feature representation can serve the few-shot learning task with more accurate features to reach a better predictive accuracy. All these feature representations are learned from upstream tasks with sufficient training classes and samples. However, we find that these extra data are not only important during the feature extraction phase, they can

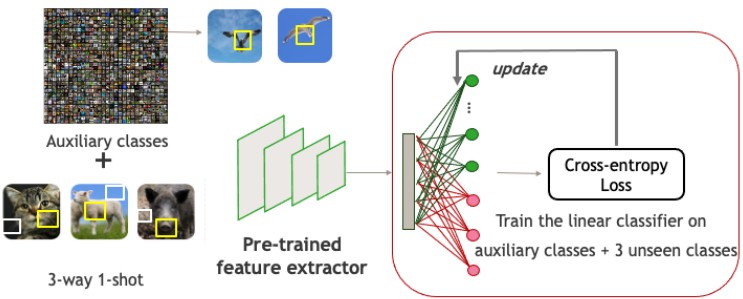

Figure 1: Adapting with auxiliary classes to downweigh irrelevant features

Table 8: Few-shot classification accuracies with auxiliary classes.

| source rep./target task | Mini-ImageNet | | | | Tiered-ImageNet | | | |
|---|---|---|---|---|---|---|---|---|
| | 5w1s | 5w5s | 10w1s | 10w5s | 5w1s | 5w5s | 10w1s | 10w5s |
| ResNetW12-cls | 61.75 | 78.98 | 47.09 | 66.93 | 71.36 | 85.56 | 58.15 | 75.77 |
| | (0.79) | (0.60) | (0.53) | (0.44) | (0.91) | (0.64) | (0.66) | (0.54) |
| ResNetW12-cls+aux | **63.70** | 79.42 | **49.29** | 67.40 | 71.92 | 85.66 | 58.59 | 76.25 |
| | (0.79) | (0.57) | (0.53) | (0.43) | (1.03) | (0.64) | (0.68) | (0.55) |
| ResNetW12-cls+rot | 63.66 | 81.68 | 49.38 | 70.48 | 72.65 | 86.17 | 59.88 | 77.25 |
| | (0.80) | (0.57) | (0.54) | (0.42) | (0.86) | (0.61) | (0.65) | (0.51) |
| ResNetW12-cls+rot+aux | **66.28** | 81.92 | **51.08** | 70.80 | **73.69** | **86.82** | 59.50 | 77.50 |
| | (0.78) | (0.55) | (0.52) | (0.41) | (0.91) | (0.60) | (0.67) | (0.53) |
| ResNetW12-cls+loc5 | 61.67 | 79.19 | 46.89 | 67.33 | 71.02 | 85.96 | 58.04 | 75.92 |
| | (0.80) | (0.59) | (0.52) | (0.41) | (0.95) | (0.63) | (0.68) | (0.53) |
| ResNetW12-cls+loc5+aux | **64.10** | **79.97** | **47.89** | 67.29 | 71.78 | 85.83 | 58.09 | 76.10 |
| | (0.75) | (0.56) | (0.52) | (0.42) | (0.93) | (0.62) | (0.69) | (0.52) |

also help to improve the performance of the final linear classifier in the adapting stage. Consider the example of a 3-way 1-shot classification of animals (sheep, cat, pig), the classifier cannot tell if "sky" is a discriminative feature for sheep if only one of the three images contain a blue sky. The classifier can confuse the features of the animal that it should learn with background features that are merely correlated with the subject.

To solve this problem, as shown in Figure 1, after determining the well-trained backbone model, we obtain the embedding representation for the few-shot support data (e.g. from a 5-way 1-shot task) as well as samples from auxiliary classes (e.g. the 64 training categories from Mini-ImageNet). Next, we train the logistic regression classifier using all these data together. In this case, the output size of the classification logits is extended (5+64). We just randomly sample 1000 examples from the training set as our auxiliary classes data. By utilizing data in auxiliary classes, our final linear classifier learns to downgrade and eliminate the contribution of those spurious features while emphasizing the unique task-related features. In our example the classifier learns to NOT associate "sky" with sheep, because the sky feature is also likely to appear in auxiliary classes examples, and such an association can lead to misclassification of the auxiliary classes examples. In the inference phase, extra connections and output logits corresponding to the auxiliary data are discarded. And the class in the current few-shot task with highest probability will be the final prediction.

The experiment result using auxiliary classes is shown in Table 8, using the best multi-task representations from the previous sections. We can see that for most models there are good improvements for 5w1s and 10w1s tasks confirming our intuition, while the improvements for 5w5s or 10w5s are smaller since there are 5 examples per class, making it less likely for the model to rely on spurious features. The improvements are also smaller for Tiered-ImageNet.

## 5.2 VOTING WITH AUXILIARY TASK INSTANCES

In Section 4 we experimented with multi-task representations. For these models, they are not just capable of predicting the target class on normal images, but also on rotated images and cropped corners of images since they are trained on these inputs. This offers yet another opportunity in improving the prediction performance via voting. We can generate a set of rotated or cropped copies

Table 9: Few-shot classification accuracies with location-based voting.

| source rep./target task | Mini-ImageNet | | | | Tiered-ImageNet | | | |
|---|---|---|---|---|---|---|---|---|
| | 5w1s | 5w5s | 10w1s | 10w5s | 5w1s | 5w5s | 10w1s | 10w5s |
| ResNetW12-cls+loc5 | 61.67 | 79.19 | 46.89 | 67.33 | 71.02 | 85.96 | 58.04 | 75.92 |
| | (0.80) | (0.59) | (0.52) | (0.41) | (0.95) | (0.63) | (0.68) | (0.53) |
| ResNetW12-cls+loc5+vote | **63.38** | **81.87** | **48.85** | **70.54** | **72.02** | 86.41 | **59.15** | **77.82** |
| | (0.77) | (0.57) | (0.54) | (0.44) | (0.93) | (0.60) | (0.69) | (0.51) |
| ResNetW12-cls+rot+loc5 | 62.56 | 80.35 | 48.43 | 68.63 | 70.91 | 85.81 | 58.11 | 75.92 |
| | (0.77) | (0.59) | (0.50) | (0.44) | (0.94) | (0.59) | (0.65) | (0.51) |
| ResNetW12-cls+rot+loc5+vote | **63.53** | **82.27** | **49.11** | **71.45** | 71.66 | **86.59** | 58.92 | **77.52** |
| | (0.82) | (0.57) | (0.55) | (0.42) | (0.95) | (0.61) | (0.66) | (0.51) |

Table 10: Few-shot classification with rotation-based voting.

| source rep./target task | Mini-ImageNet | | | | Tiered-ImageNet | | | |
|---|---|---|---|---|---|---|---|---|
| | 5w1s | 5w5s | 10w1s | 10w5s | 5w1s | 5w5s | 10w1s | 10w5s |
| ResNetW12-cls+rot | 63.66 | 81.68 | 49.38 | 70.48 | 72.65 | 86.17 | 59.88 | 77.25 |
| | (0.80) | (0.57) | (0.54) | (0.42) | (0.86) | (0.61) | (0.65) | (0.51) |
| ResNetW12-cls+rot+vote | 62.89 | 81.32 | 48.91 | 70.07 | 73.02 | 86.70 | 59.76 | 77.84 |
| | (0.81) | (0.57) | (0.53) | (0.42) | (0.91) | (0.62) | (0.66) | (0.52) |

of the input image, and run the target classifier on all of them, and take the majority prediction as the final prediction. In the following we consider two voting schemes, one based on generating 4 copies of rotated input image for voting, and another one based on generating the 5 copies (4 cropped corners and one rescaled input) for voting, corresponding to the rotation prediction and location prediction task we used.

We can see from Tables 9 and 10 the results of voting. Voting based on the cropped images from the location prediction problem is effective in improving the prediction accuracies across all the various few-shot learning tasks. The accuracies improve on average by 2%, which is quite large for these problems. On the other hand, voting based on rotated versions of the same image does not seem to help, presumably because unlike cropping the rotated images all contain the same information about the class. It degrades performance slightly on Mini-ImageNet but slightly improves performance on Tiered-ImageNet, but both are within standard error. We also tried voting based on rotated and cropped copies using the base classification representation. But this degrades the performance since the classification model was not trained on classifying cropped or rotated images, indicating the importance of multi-task training.

We finally compare our multi-task representations with the above feature selection and voting heuristics against some of the state-of-art methods based on inductive learning published in the past two years, and the results are shown in Table 11. The results on 5w1s and 5w5s learning are shown as they are the most commonly published numbers. We use the cls+rot+loc5 representation with auxiliary classes and location voting as our best model for Mini-ImageNet, and cls+rot with auxiliary classes as our best model for Tiered-ImageNet. Our approach is competitive with these state-of-art models. Our best results are obtained with the original input image, as it supports the cropping of

Table 11: Comparison of our approach against some state-of-art methods.

| Method | Backbone | Mini-ImageNet | | Tiered-ImageNet | |
|---|---|---|---|---|---|
| | | 5w1s | 5w5s | 5w1s | 5w5s |
| DeepEMD(Zhang et al. (2020)) | ResNet12 | 65.91(0.82) | 82.41(0.56) | 71.16(0.87) | **86.03**(0.58) |
| FEAT(Ye et al. (2020)) | ResNet12 | **66.78**(0.20) | 82.05(0.14) | 70.80(0.23) | 84.38(0.16) |
| MetaOptNet(Lee et al. (2019)) | ResNet12 | 62.64(0.61) | 78.63(0.46) | 65.99(0.72) | 81.56(0.53) |
| MTL(Sun et al. (2020)) | ResNet12 | 61.20(1.80) | 75.50(0.80) | 65.60(1.80) | 80.80(0.80) |
| CAN(Hou et al. (2019)) | ResNet-12 | 63.85(0.48) | 79.44(0.34) | 69.89(0.51) | 84.23(0.37) |
| Distillation(Tian et al. (2020)) | ResNet-12 | 64.82(0.60) | 82.14(0.43) | 71.52(0.69) | **86.03**(0.49) |
| Boosting(Gidaris et al. (2019)) | WRN-28-10 | 63.77(0.45) | 80.70(0.33) | 70.53(0.51) | 84.98(0.36) |
| Fine-tuning(Dhillon et al. (2019)) | WRN-28-10 | 57.73(0.62) | 78.17(0.49) | 66.58(0.70) | 85.55(0.48) |
| Centroid align(Afrasiyabi et al. (2020)) | WRN-28-10 | 65.92(0.60) | 82.85(0.55) | **74.40**(0.68) | **86.61**(0.59) |
| Ours (best model, crop on original) | ResNetW12 | 65.23(0.81) | **83.37**(0.56) | 73.69(0.91) | **86.82**(0.60) |
| Ours (best model, 84x84 resized) | ResNetW12 | 63.05(0.79) | 80.11(0.57) | 69.78(0.88) | 84.86(0.65) |

184x184 images without upscaling for the location prediction task. We also show results on the 84x84 resized image, using cls+rot models for both Mini-ImageNet and Tiered-ImageNet since we cannot include the location prediction task with this resolution. There is a considerable gap between these two input preprocessing approaches, which we discuss in Appendix A.1. We believe the biggest benefit of our approach is the ability to deal with two concerns in few-shot learning separately: transfering to a new domain and dealing with the small number of examples given in that new domain. We handle the first one by studying representations for few-shot learning transfer, and this problem can benefit from the numerous works done in representation learning in computer vision and NLP in recent years. We handle the second problem by imposing extra constraints when adapting a linear classifier through prior knowledge. This modular approach can allow progress on solving either problems translate to improvements in few-shot learning.

## 6 FUTURE WORK AND CONCLUSIONS

We have presented in this work a systematic study of different learned feature representations for few-shot learning. Coupled with some new heuristics for feature selection, we find that this transfer learning approach with multi-task representations is competitive with state-of-art methods. For future work we would like to study if deeper networks would allow us to learn these multi-task representations better, and also obtain a better understanding of how features are shared or compete against each other in these multiple tasks. We are also interested in exploring what other types of structures or constraints can be exploited during the classifier adaptation stage.

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

## A  APPENDIX

### A.1  EFFECT OF INPUT IMAGE SIZE ON FEW-SHOT CLASSIFICATION ACCURACY

In the literature the datasets Mini-ImageNet and Tiered-ImageNet are usually constructed from the standard ImageNet data(Deng et al., 2009), but depending on the interpretation of the input image size of 84x84 from the original papers(Vinyals et al., 2016; Ren et al., 2018), there could be two different versions of Mini-ImageNet or Tiered-ImageNet constructed. One approach directly resizes the input image file to 84x84, while the other approach performs random resized crops of size 84x84 analogous to the common data augmentation approach in ImageNet training (crops of 224x224 instead). We have seen implementations of data loaders that read from resized 84x84 python pickle files, and also data loaders that directly perform random resized crops from the input jpeg image.

Table 12: Few-shot classification on different input image resolution.

| source rep./target task | Mini-ImageNet | | | | Tiered-ImageNet | | | |
|---|---|---|---|---|---|---|---|---|
| | 5w1s | 5w5s | 10w1s | 10w5s | 5w1s | 5w5s | 10w1s | 10w5s |
| ResNetW12-cls(84x84 resized) | 60.53 (0.80) | 78.44 (0.61) | 45.77 (0.53) | 65.87 (0.43) | 68.38 (0.94) | 83.25 (0.67) | 54.83 (0.66) | 73.02 (0.53) |
| ResNetW12-cls(84x84 random crop) | 61.75 (0.79) | 78.98 (0.60) | 47.09 (0.53) | 66.93 (0.44) | 71.36 (0.91) | 85.56 (0.64) | 58.15 (0.66) | 75.77 (0.54) |

Table 13: Few-shot classification using different number of augmented examples from training set.

| source rep./target task | Mini-ImageNet | | | |
|---|---|---|---|---|
| | 5w1s | 5w5s | 10w1s | 10w5s |
| ResNetW12-cls | 61.75(0.79) | 78.98(0.60) | 47.09(0.53) | 66.93(0.44) |
| ResNetW12-cls-aux1 | 63.62(0.77) | 79.53(0.58) | 47.97(0.50) | 67.06(0.44) |
| ResNetW12-cls-aux5 | 63.96(0.81) | 79.39(0.58) | 48.17(0.53) | 67.16(0.43) |
| ResNetW12-cls-aux10 | 63.69(0.83) | 79.43(0.60) | 47.90(0.55) | 67.31(0.43) |

But these two versions of the data contain different amount of information as images in the second approach is of higher resolution, especially when data augmentation is used. Data augmentation is used on the support set for both logistic regression and MAML. Table 12 shows the few-shot transfer accuracy of the base classification model on Mini-ImageNet and Tiered-ImageNet. In general using the original ImageNet input image compared to the downsampled 84x84 version gives 1-2% advantage for Mini-ImageNet, and a 2-3% advantage for Tiered-ImageNet, for the models we considered in this paper. In this work we focus on the original input image, because representation learning methods like location prediction needs a larger 184x184 input image to avoid effect of interpolation in upscaling, and constrative learning benefits from having a larger set of random crops from a higher resolution image. This is consistent with the usual training of representation learning methods in ImageNet because the input images are not resized to 224x224 to allow for more types of input transformations. We discovered this issue of input resolution when we downloaded two different versions of the data and noticed there are considerable differences in their accuracies.

## A.2 Effect of Number of Augmented Auxiliary Examples on Few-Shot Classification Accuracy

We also consider a more systematic scheme for augmenting the support set with auxiliary classes from the training data apart from randomly sampling 1000 examples from the training set. In Table 13 we consider augmenting 1, 5 and 10 examples per training class(64 classes) to the support set on Mini-ImageNet. We can see that the improvement is similar for different number of examples augmented. One might be able to obtain better results by increasing the number of augmented examples but at the same time control their total sample weight relative to examples in the support set, but we do not explore this further in this work.

