# OpenReview forum: "A Study on Representation Transfer for Few-Shot Learning"
_ICLR.cc/2022/Conference — ICLR 2022 Submitted_

### Official Review · Reviewer_28ox · 2021-10-28

**Correctness:** 3
**Technical Novelty And Significance:** 3
**Empirical Novelty And Significance:** 3
**Recommendation:** 5
**Confidence:** 5

**Main Review:**

# Main review
The main part of the paper is dedicated to the study on the various training approaches for the transferred backbone. This study confirm the observations already published in the literature (here work of Tian et al is especially cited). Author could also have cited work of Mangla et al that also addressed this point and provide some insights on the benefits of using additional self-supervised task in the context of Few Shot Learning. This study part is quite long and some experiments could be omitted (e.g. training the backbone only on one self-supervised task) and does not really bring some new learnings with respects to previous works. In previous works it is already clear that self-supervised tasks should be considered as additional tasks.

Note also that contrastive loss (from Khosla et al) has also been proposed recently in the context of Few Shot Learning by Ma et al, where 63.76% accuracy is reported on 5w1s (see table 3 of the paper)  which is rather pointing in favour of using such loss in the context of Few Shot Learning. This is contradictory to the comment made on table 4. This is not to point something to be cited, but rather to highlight the weakness of the observations and conclusions made in this part.

Although being cited, no comparison result to Tian et al is provided in the benchmarks.

Section 5 presents some really interesting tricks to improve the classifier.
From table 8, a significant systematic gain is observed when using auxiliary classes. I find it as being the main contribution of this paper. Additional studies would be interesting to be performed in order to see what is the impact of the number of additional samples used as well as more detailed explanation on how additional samples are used.
- It is not clear here if all of the 1000 additional samples are always used here  (1000 is not a multiple of 64)
- does the 64 additional classes are always used?
- what is the impact if using less auxiliary classes with less samples per class?

In the final comparison, a combination of the various approaches is compared to SOTA solutions. However it is not clear from where the benefits comes from in the selected combinations. Especially since we can observe that the combination is different for mini-ImageNet and tiered-ImageNet. Could we have more insights why the voting solution does not work for tiered-ImageNet?

Table 11 reports comparative results with other SOTA approaches. However since backbone architecture is different and with the use of auxiliary classes and voting mechanism it is not clear where the benefit come from. Ablation studies is required to see what are the relative benefits of the varying tricks. Especially when we can observe that the settings between mini-ImageNet and tiered-ImageNet are different (location voting used only for mini-ImageNet)


# Minor remarks
- page 3, it is mentioned from results in Tables 1 and 2, that it is better to train on 10-ways tasks. But from those tables 5w5s and and 5w1s are still competitive for imagenet. conclusions here are not that straightforward.
- table 3 is not cited in the paper. It should be cited in section 3.2 when discussing about MAML based training vs supervised training.


# Additional references
- Mangla et al, "Charting the right manifold: Manifold mixup for few shot learning"; WACV 2020.
- Prannay Khosla, Piotr Teterwak, Chen Wang, Aaron Sarna,Yonglong Tian, Phillip Isola, Aaron Maschinot, Ce Liu, andDilip  Krishnan.Supervised  contrastive  learning.arXivpreprint arXiv:2004.11362, 2020
- Jiawei Ma, Hanchen Xie, Guangxing Han, Shih-Fu Chang, Aram Galstyan, Wael Abd-Almageed, Partner-Assisted Learning for Few-Shot Image Classification. arxiv:2109.07607, 2021

**Summary Of The Paper:**

The paper provides a study on different approaches to train a backbone network used in a transfer fashion for Few Shot Learning.
Representation learned from MAML, supervised classification, and self-supervised tasks are considered here and their performance are compared. From this analysis it is confirmed that the use of multi-task trained representation (based on supervised classification and self-supervised auxilary tasks) leads to the most performing backbone.

Furthermore it is proposed to improve the classifier part using two additional tricks:
- use of auxiliary classes to help eliminate irrelevant features
- use of voting with image transformations (rotation, location splitting)
This additionnal trick allows to boost performance over traditionnal simple classifier.

Final results show performance on par with various state of the art solutions.

**Summary Of The Review:**

From the contributions reported by the authors:
- contribution 1 and 2 are not really novel and do not really provide some new insight with respect to previously published works and current trends in training a backbone for Few Shot Learning
- contribution 3 (feature selection, voting) is quite novel and provide some interesting gains, but more experiments would be needed to assess the relative benefits of those proposals.

Considering that the focus in this paper is more put on contribution 1 and 2, a significant reorganization of the paper as well as more experimental results should be done in order to focus more on contribution 3. That's why my vote is to reject this paper.

---

### Official Review · Reviewer_Tk8a · 2021-10-31

**Correctness:** 3
**Technical Novelty And Significance:** 2
**Empirical Novelty And Significance:** 2
**Recommendation:** 5
**Confidence:** 4

**Main Review:**

Studying the utility of different representations for few-shot learning tasks is an interesting and important topic, due to the success and wide adoption of the transfer learning paradigm. Jointly training for different objectives is also interesting to study, as well as the proposed voting and auxiliary class downstream training methods proposed here. These have the potential to produce impactful results. The paper is also clearly written for the most part and easy to follow. However, the study is limited to simple datasets, and the proposed methods do not yield a consistent gain, especially on tiered-ImageNet. Some additional concerns and feedback below:

Correctness
- “The paper Tian et al. (2020) show that by just pre-training … and adapting using logistic regression, they can obtain results … that are competitive with state-of-the-art algorithms” - This isn’t exactly true. While the approach of Tian et al indeed doesn’t use meta-learning, their approach is more complicated than what this sentence makes it sound, as it involves a distillation phase aside from just standard pre-training.

- ‘... and train a logistic regression model on top of it during the adaptation stage on the query set’ - this training should take place on the support set, not the query set. I assume this is just a typo but it would be great if the authors could clarify.

- ‘Transfer learning, on the other hand, … , a much simpler task model (e.g. a linear classifier) instead of having to learn feature extraction at test time’. This doesn’t seem correct - meta-learning approaches sometimes learn simple models too, e.g. Prototypical Networks is a form of a linear classifier, and transfer learning often involves fine-tuning the entire network at test time [1] (some examples in the context of few-shot classification: [3, 4]). The fact that just training a linear classifier on top of pre-trained features works so well for mini- and tiered- imagenet might just be a symptom of the simplicity of these datasets and resulting transfer problems, but this is not true of transfer learning more generally.

- ‘it is very difficult to figure out which feature is relevant given only a handful of examples’ - is there some evidence to support this claim? Doesn’t the success of simple linear classification on top of pre-trained features contradict this claim to some degree?

Relationship to previous work
- The observation that using a larger ‘way’ during training than testing has already been made in prior literature [4]. The authors there also hypothesized that this is due to harder training tasks being beneficial for representation learning. Though in this case the experiments are done with MAML instead of Prototypical Networks, this connection should still be discussed.

- In addition, the effect of the number of shots used at training time on test performance on different configurations of tasks has been explored both theoretically and empirically in the past [5,6].

- The approach used here is reminiscent of the ‘input harmonization’ in [7]. Can we get an additional performance boost from also incorporating the lasso architecture proposed there? This allows different tasks to have slightly different weights instead of forcing them all to use exactly the same feature extractor.

- [8] (and approaches cited within) also study transfer learning with several different types of pre-trained representations, on several different data-limited downstream tasks. They explore both supervised and self-supervised representation learning approaches. It is worth discussing the connections to that work and how the goals of this work differ from previous studies.

Additional experiments
- What happens if a more naive approach is used for multi-task training, where each task receives different inputs (e.g. the classification loss is computed on the original images instead of the rotated or cropped ones). The assumption made here is that this would perform worse, but it would be useful to actually report these results and verify this hypothesis. It seems, for instance, that [9] performed joint training between a supervised a self-supervised objective without making any modifications, and still observed improvement in performance.
- Can location-based voting and rotation-based voting be combined? More generally, how does this compare to voting with random augmentations (e.g. those typically used for data augmentation, like color jittering, horizontal flipping, etc)? This would help understand if this voting is beneficial only when the augmentations chosen are the same ones used for training.
- The authors hypothesize that the reason auxiliary class training helps is to alleviate the issue of learning spurious features. Is there some evidence to support this hypothesis? Perhaps looking at the gap between the support and query accuracy in both cases (with and without the auxiliary classes) would be helpful, since if the classifier is learning spurious features perhaps this would lead to overfitting on the support set and not generalizing to the query set?

Clarity
- It’s not entirely clear how logistic regression differs from fast adaptation of MAML, when only adapting the final layer of MAML. I’m assuming the main difference is that a new random readout layer is initialized instead of re-using MAML’s meta-learned classification layer. Is there also a difference in the number of steps, e.g. logistic regression takes more steps than fast adaptation? It would be good to clarify exactly what the differences are.

- ‘To further simplify the re-initialization step’, citing the Reptile paper. This sentence is confusing; it’s not clear what re-initialization step means and how this paper simplifies it. It seems more appropriate to mention Reptile in the context of first-order approximations for MAML.

- The term ‘domain’ is used in an inappropriate way throughout the paper. I wouldn’t say that the test set of mini-ImageNet comprises a different domain than the training set of mini-ImageNet. Referring to it as ‘held-out class split’, for instance, would be more appropriate.

- Why not just refer to ‘modified MAML’ by its name, ANIL (from Raghu et al, as cited).

References
- [1] Do Better ImageNet Models Transfer Better? Kornblith et al. CVPR 2019.
- [2] Meta-Dataset: A Dataset of Datasets for Learning to Learn from Few Examples. Triantafillou et al. ICLR 2020.
- [3] Comparing Transfer and Meta Learning Approaches on a Unified Few-Shot Classification Benchmark. Dumoulin et al. NeurIPS 2021.
- [4] Prototypical Networks for Few-shot Learning. Snell et al. NeurIPS 2017.
- [5] A Theoretical Analysis of the Number of Shots in Few-shot Learning. Cao et al. ICLR 2020.
- [6] Learning Flexible Classifiers with Shot-CONditional Episodic (SCONE) Training. Triantafillou et al. Meta-learn workshop, NeurIPS 2020.
- [7] Multi-task Self-Supervised Visual Learning. Doersch et al. ICCV 2017.
- [8] A Large-scale Study of Representation Learning with the Visual Task Adaptation Benchmark. Zhai et al. 2020.
- [9] Boosting Few-Shot Visual Learning with Self-Supervision. Gidaris et al. 2019.



**Summary Of The Paper:**

This paper conducts an empirical study on the usefulness of different representations for downstream few-shot learning tasks, using the transfer strategy of training a new linear layer on top of the frozen pre-trained network. They also propose to utilize auxiliary data during this downstream learning phase: instead of learning an N-way readout layer for an N-way classification task, they learn a (N+T)-way one, where T is the number of training classes, using a subset of the training dataset for this. To perform inference on the query set, though, they only consider the N classes appearing in the given task. They hypothesize that this approach prevents the linear classifier from paying attention to spurious features, and it sometimes yields a gain in performance. They also propose to combine different training objectives, including the supervised classification objective and different self-supervised ones in a multi-task manner, and propose to do so by keeping the input the same for all objectives involved, meaning that to train the supervised loss together with the rotation, the former will also be computed on rotated images. Finally, they propose a voting scheme, where the predictions for different augmentations of each input image are aggregated to form the final prediction. All experiments are conducted on mini-ImageNet and tiered-ImageNet, and gains are reported mostly on the former.


**Summary Of The Review:**

Although the paper studies an interesting topic, I have some concerns about correctness of various statements in the paper, the relationship with prior work that has explored similar questions, and the need for additional experiments and analyses to aid in better understanding where the observed performance gains come from. Finally, a weakness of this work is that the study is limited to simple datasets and a linear transfer learning adaptation technique, which makes me question whether the conclusions will generalize to other scenarios.

---

### Official Review · Reviewer_G5x6 · 2021-11-01

**Correctness:** 3
**Technical Novelty And Significance:** 1
**Empirical Novelty And Significance:** 2
**Recommendation:** 3
**Confidence:** 5

**Main Review:**

The paper is generally well written with a clear explanations and research direction.
The topic is relevant and of interest to the community.

After initial description I was saddened that the analysis of feature representation of few-shot methods involved only MAML. It is in fact significant method that influenced many other, but its performance was quickly surpassed by more advanced methods which should be analyzed instead. At least some more methods should be analyzed as then instead of studying few-shot methods the paper studies MAML only (which might mean that conclusions might not generalize). The self-supervised methods studied (rotation, location and contrastive) are also very simple, and although their analysis is good, the proposed way of combining them is difficult to see as a strong technical novelty. The same goes for the voting system built on top of the proposed training scheme. The results confirm the importance of both, however the technical contribution itself seems weak.

In Table 11, only the authors method is using modified ResNetW12 -- this raises the question how significant that is? If the authors' method requires it, then other methods should be compared fairly with same backbone. Regardless of this, the results do not build a strong case for the proposed method (since only for a 5w5s on miniImageNet it outperforms existing methods). Comparable results would be impressive if the proposed method would provide some additional benefit, however none were mentioned.

**Summary Of The Paper:**

The authors study impact of self-supervised methods on feature representation in few-shot learning task. They analyze the way features are transferred for MAML method and 3 self-supervised methods. They introduced a method combining self-supervised representation training methods with voting system in order to improve performance.

**Summary Of The Review:**

Interesting case study of feature representations for few-shot learning. However, it lacks depth (only MAML analyzed as few-shot method) and the proposed solution lacks technical novelty and convincing results.

---

### Official Review · Reviewer_63j7 · 2021-11-02

**Correctness:** 3
**Technical Novelty And Significance:** 2
**Empirical Novelty And Significance:** 3
**Recommendation:** 6
**Confidence:** 4

**Main Review:**

* Although the ideas of using the MAML algorithm to adapt the last layer of the neural network which performs very close to adapting the whole network (Raghu et al., 2019) and simple baselines perform close to SOTA from learning a good representation through a proxy task (Tian et al., 2020) already exist. The authors of the paper acknowledge it and conducted experiments on how simple these baselines (ConvNet, ResNet) can be. And which proxy tasks (rotation prediction, location prediction) are helpful.
* In-depth analysis and experimentation with various techniques, architectures like ResNet, ConvNet, and sizes of the network help point out the limitations of those techniques and lead to the critical finding that complex tasks (combining classification, rotation prediction, and location prediction) tend to give better representations for few-shot classification.
* The use of transfer learning with multi-task representations and novel tricks like “eliminating irrelevant features using auxiliary classes” improved the performance on 5w1s, 5w5s, 10w1s by emphasizing the unique task-related features. The other trick is “voting with auxiliary task instances” where a set of rotated or cropped copies of the input image are given with the input image may not have any impact on the learned feature representations but it helped to improve the accuracy even closer to SOTA.
* The description of various hyperparameters and the reasoning for constraints used in the experiments helped to better understand and compare the results to other methods.

#### __Concerns:__

I have some queries about the studies presented in the paper.

* In Section 3.2, Imagenet pre-training, It’s unclear how much data is used?
    * As miniImagenet, TieredImageNet are subsets of ImageNet, Is data of common classes removed from the pre-training data at the time of pre-training?.
    * To what extent does the amount of data and the diversity of data used in the pre-training affect the representations and performance of the various models.
* The authors have chosen the representation size to be 640, may I know the reason behind it? Will the performance of the various models used in the studies vary with size?
* Although the paper provides various experiments on the Initialization based method (MAML), Self Supervised Methods, will the proposed tricks improve Metric Based Models like Prototypical Networks, Matching Networks?

#### __Other suggestions:__
 It is difficult to follow section 3.2 due to the missing reference to Table 3.



**Summary Of The Paper:**

This paper presents an interesting study comparing the various feature representations for few-shot classification tasks (in Computer Vision) learned from the supervised classification with/without MAML, from multi-task prediction, and self-supervised tasks like rotation prediction and location prediction, and contrastive learning.

The paper also introduces two tricks “irrelevant feature elimination using auxiliary classes”  and “voting with auxiliary task instances” which made the ImageNet pre-trained model’s performance on downstream tasks of 5 way 1 shot & 5 way 5 shot close to SOTA (State Of The Art) while handling the issue of a small sample size of 5 examples per class.

**Summary Of The Review:**

The idea of improving existing baselines to be almost comparable to SOTA with some techniques may not be a novel one but this paper stands out in experimenting and in-depth analysis of the various techniques and also combining them with heuristic tricks like feature selection and voting to handle the issue of small sample size. Hopefully, the authors can address my queries/concerns (mentioned above) in the rebuttal period.

---

### Official Review · Reviewer_EEwV · 2021-11-03

**Correctness:** 3
**Technical Novelty And Significance:** 1
**Empirical Novelty And Significance:** 2
**Recommendation:** 3
**Confidence:** 4

**Main Review:**

My main concern of this paper is its novelty. In my opinion, it is already a kind of common sense that training with more complex data results in better feature representation. (that's why the race to develop a giant pre-learning model with big data is on)
Using multiple source tasks is also a straightforward direction to go. For example, it is well-studied in the following paper.
  - Zamir et al., Taskonomy: Disentangling Task Transfer Learning, CVPR 2018.

I understand that the focus of this paper is different (i.e., few-shot, self-supervised tasks),  but at high level, the findings and conclusions in the paper seem rather obvious to me. Or, is this result really surprising for few-shot learning? If so, please clarify it as the reviewer doesn't understand the point.

As for the feature selection (auxiliary class) and voting methods, I can find no novelty in them either. In my understanding, they are nothing but multi-task learning and test-time augmentation, respectively.

**Summary Of The Paper:**

This paper studies few-shot learning from the viewpoint of transferring feature representations. The authors investigate few-shot performance with varying complexities of source tasks together with a few empirical tricks for improvements. The main finding in the paper is that transferring from more complex source tasks tend to result in better performance. Accordingly, using multiple source tasks is also found to be useful.

Strengths:
(+) The paper is clearly motivated and well-written.
(+) Reported scores could be useful to server as a baseline in this direction of research.

Weaknesses:
(-) No technical novelty.
(-) Results and conclusions seem rather obvious.

**Summary Of The Review:**

Overall, I find this paper presents an interesting case study of feature transfer focusing on few-shot learning, it lacks enough novel insights and technical contributions to warrant publication at ICLR.

---

### Decision · Program_Chairs · 2022-01-20

**Decision:**

Reject

**Comment:**

This work studies a number of feature representations for few-shot classification, including representations learned from MAML, supervised classification, and some self-supervised tasks. The main conclusion of the study is that learning from more complex tasks result in better representations for few-shot classification. As a practical solution, then, the authors suggest using representations learned from multiple tasks for few-shot classification.

The paper studies an important problem in machine learning, and reviewers all appreciate that. However, they raised concerns about the draft in its current state. Authors replied to these comments, and while reviewers acknowledged and appreciated the responses and the revision of the draft, unfortunately that did not convince the reviewers. Several major concerns remained unresolved at the end. Specifically, EEwV believes that the paper is a case study, which though useful, does not bring deep new insights or findings. 63j7 believes that even after revisions made to the paper, there are additional experiments required to understand and examine the claims. Tk8a finds the submission unready for publication due to weak experimental analysis and suggests running additional experiments to examine the hypotheses made by the authors (e.g, relating to spurious features, the need of input harmonization, the benefit of voting) and better tie in the findings of this work to related work. Tk8a provides a list of concrete suggestion along these lines. Similar to EEwV, 28ox also thinks that the paper lack novelty or does not really bring new insights on the way to train the backbone.

Based on these comments, and the ratings, I encourage authors to address these issues and resubmit.